# Natural Dyes Used as Organic Coatings UV Protecting for Food Packages

Tersilla Virgili [1,*], Mariacecilia Pasini [2,*], Michele Guizzardi [1], Negar Tizro [2] and Monica Bollani [1,*]

1   Institute for Photonics and Nanotechnologies (IFN), National Research Council (CNR), Piazza Leonardo da Vinci 32, 20133 Milan, Italy; michele.guizzardi@polimi.it
2   Institute of Chemical Sciences and Technologies (SCITEC), National Research Council (CNR), Via Corti, 20133 Milan, Italy; negartizro@gmail.com
*   Correspondence: tersilla.virgili@polimi.it (T.V.); mariacecilia.pasini@scitec.cnr.it (M.P.); monica.bollani@ifn.cnr.it (M.B.)

**Abstract:** Nowadays, the use of biodegradable and bio-derived plastics such as poly(lactic acid) (PLA) and cellulose in food packaging applications are replacing the use of different and more conventional oil-derived polymers that are much more expensive and unsuitable for the environment. However, their high transparency to ultraviolet (UV) radiation limits their current commercialization. Therefore, this study focuses on the deposition of organic thin films on commercial PLA and cellulose in order to enhance their performance, in particular for UV shielding. Coatings with different natural and biocompatible (edible) molecules, quinine and p-Coumaric acid, dispersed in different matrices are deposited by comparing different deposition techniques such as spray coating and spin coating. Morphological characterizations are carried out with atomic force microscopy (AFM) and scanning electron microscopy (SEM), with spectroscopic characterizations performed by light transmission measurements. Our results show that it is possible to enhance the UV protection of packaging using a suitable deposition with these biocompatible materials.

**Keywords:** food packaging; UV protection; organic coatings; PLA; cellulose; quinine; p-Coumaric acid





## 1. Introduction

Food packaging is used to protect food from environmental contamination such as light, humidity, physical damage, etc. [1]. The use of the right packaging can be the key to avoiding food wastage and losses. There have been many recent developments made in packaging technology to reach smart and active packaging [2]. Biodegradable polymers, and thus green packaging, are viewed as a feasible strategy to reduce the serious environmental problem caused by the excessive use of conventional plastic polymers as packaging materials [3]. In addition to cheaper raw material costs, the process of making starch-based plastics requires less overall energy compared to the process of creating petroleum-based ones [1]. They also releases fewer greenhouse gases during manufacturing, which is another environmentally friendly benefit [1]. Among the possible biopolymers candidates, poly(lactic acid) (PLA) [4,5] and cellulose [6,7] possess the highest commercial potential due to their abundant renewable natural resources, biocompatibility, and transparency, plus their mechanical properties are comparable with those of conventional polymers [8–10].

However, several major limitations restrict the application of bioplastics in food packaging materials, including their high cost compared to conventional plastics, brittleness, thermal instability, low melt strength, difficult heat sealability, high water vapor, high oxygen permeability, bad processability, and poor impact resistance [11,12]. To improve the properties of bioplastics (especially their barrier capacities toward gases and water) and to extend the shelf life of products, different strategies and techniques have been investigated. For example, the effects of whey protein isolate-based film incorporated with essential oils [13], the use of coating biobased films, incorporation of nanoparticles

or biopolymer cellulose, and chemical/physical modification, such as crosslinking, were studied [14] However, bioplastics' permeable nature to gas, along with their transparency to UV radiation, restrict their use for several applications, especially related to food packaging [15,16]. Pedroni et al. [3] recently studied PLA coated with $WO_x$ thin films and they showed improved properties, including serving as a barrier to oxygen, UV light protection, and antimicrobial activity. Most of the previous works are related to coatings with nanostructures. In this paper, we take a different approach, using natural dyes blended with biocompatible optically inert matrices as organic coatings. We show the possibility of reducing their UV transparency after smart coatings with commercial and green materials such as the p-Coumaric acid and quinine. We use different laboratory surface coating methods such as spray and spin coating. As a non-thermal technology, surface coating is more promising than molding or extrusion in the production of food packaging containing heat-sensitive active molecules. Furthermore, it also has advantages over direct incorporation to preserve bulk properties of packaging materials, including mechanical and physical properties. Indeed, spin coating is a technique used to deposit coating materials on a flat substrate through centrifugal force: the coating material is applied onto the center of a substrate and then distributed evenly on the substrate through spinning at a high velocity. While spin coating is generally a laboratory technique that can only be used for small-sized substrates, spray coating is a technique generally used to deposit the coating solution or suspension onto the substrate on both an industrial and laboratory scale [17]. In this process, the liquid mixture used for the coating is atomized into droplets, then transferred to the substrate, with the formation of the coating obtained after the deposition of the droplets and subsequent evaporation of the solvent on the surface. Moreover, spray coating as reported is a simple, low-cost technique [18] with the ease to scale up for industrial production. The coated films are studied morphologically to confirm that the surface quality of the packaging films is preserved after the deposition treatments. Spectroscopic measurements instead show the possibility to increase the UV barrier of the commercial packaging materials using specific polymeric coatings, predicting their future use in the market.

In order to create a suitable packaging for food, all materials used, such as active molecules, polymeric hosts, and solvents, are chosen so as to be suitable for contact with food. The two UV-absorbing chromophores chosen are quinine and p-Coumaric acid, with both dyes having strong absorption in the UV region [19,20]. Quinine has been known for some time as the basis of the first antimalarial drugs and also the preparation of beverages such as tonic water, and therefore is not only suitable for contact with food but for use in food preparations [21]. Furthermore, since quinine is derived from the stem of the cinchona (quina-quina) tree, it can be fully considered a sustainable material. Quinine can perfectly absorb the UV region from 200 to approximately 370 nm and has good solubility in water.

P-Coumaric acid can be found in a wide variety of edible plants and fungi such as peanuts, navy beans, tomatoes, carrots, basil, and garlic, as well as in wine and vinegar. It is also found in barley grain, so it can be considered as an "edible dye". According to recent evidence about the beneficial effects of p-Coumaric acid, nephropathies, cardiovascular diseases, neuro-inflammatory diseases, liver diseases, cancers, and some metabolic disorders could potentially be controlled by this natural herb [22]. P-Coumaric acid is soluble in alcohol as ethanol and is transparent in the visible region, absorbing in the UV region from 200 nm to ~370 nm

## 2. Materials and Methods

### 2.1. Materials

Two different substrate films based on cellulose (C) and poly(lactic acid) (PLA) are provided by Corapack Srl (Brenna CO, Italy) in the framework of the Regione Lombardia Spatial project. Quinine hydrochloride dehydrate (Q), p-Coumaric acid (PCA), and polyvinyl alcohol (PVA Mw reported 30,000–70,000) are purchased by Aldrich (St. Louis,

MO, USA). The polymeric host polyvinyl butyrate derivative (PVB, Mowital BH®) is purchased by Kuraray (Hattersheim am Main, Germany).

### 2.2. Coatings Preparation

Film preparation: 2 cm × 2 cm films of PLA and C are fixed on a glass slide with scotch tape. A first solution PVA:Quinine (PVA-Q) in water with the ratio 60 mg:15 mg PVA:Q for each mL of $H_2O$ is used (approximately 25% dye with respect to the polymeric host). This solution is used for both spray- and spin-coating deposition. A second solution labeled PVB: p-Coumaric acid (PVB-PCA) is prepared stirring the PVB and PCA in ethanol with the ratio (60:6) mg for each mL (approximately 10% dye with respect to the polymeric host). The same PVB-PCA mixture is used for both spray- and spin-coating deposition. In the case of spin coating, each sample is coated twice by 200 μL at 1000 rpm for 120 s. After this time, the solvent is completely evaporated. The solution for the spray coating, instead, is transferred to a spray jar and sprayed on the substrate in horizontal form. After the spray coating, the samples prepared using ethanol are left at room temperature for about 1 h to ensure complete evaporation of the solvent. In the case of water as a solvent, the samples are left for about 2 h.

Figure 1 shows the different coating techniques (panel a) and the chemical structures of the organic molecules used (panel b) and of the polymeric hosts (panel c).

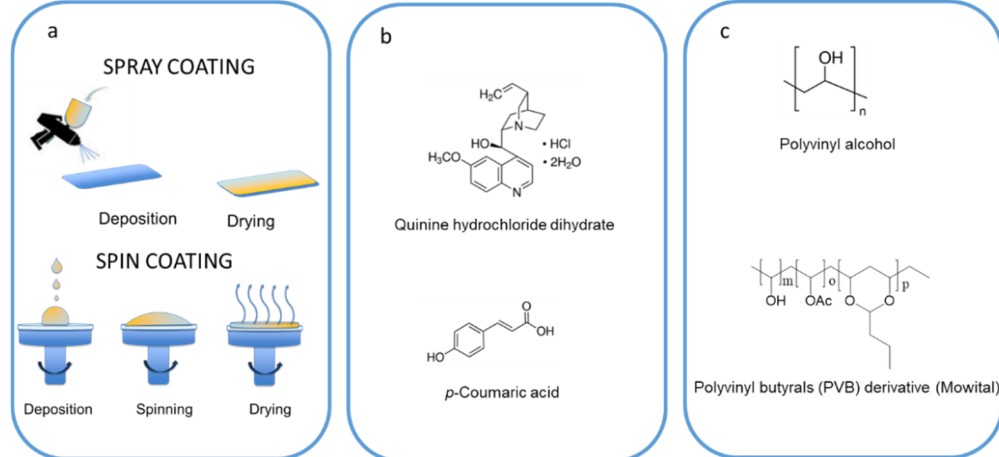

**Figure 1.** (**a**) Deposition techniques; (**b**) Active materials; (**c**) Polymeric hosts.

### 2.3. Samples Characterization

We systematically address the performance of the coated flexible food packaging films by a combination of atomic force microscopy (AFM Innova of Veeco company ,Binasco, Italy) and scanning electron microscope (SEM, Philips XL30 S-FEG SEM, Milan, Italy) tools. AFM topography images are acquired in tapping mode with a Veeco Innova instrument. Super-sharp silicon probes (typical radius of curvature 2 nm) are used for high-resolution imaging of the defects in the organic coating, while standard silicon probes are employed for large-area scans in order to evaluate the sample roughness. The root mean square (rms) roughness is calculated as the standard deviation of the topography by Gwyddion software, as descripted in [23]. PLA- and C-coated samples are observed using a Philips XL30 SEM at different magnifications: the micrographs are acquired using a secondary electron detector (5 kV acceleration voltage), probe current of 0.20 nA and working distance of 4.2 mm. Pieces of coated film (3 cm × 3 cm) are mounted directly on a rectangular metallic sample holder, observed in planar, and tilted.

Optical properties are studied using a Spectrophotometer UV-VIS-NIR (V-570, Jasco, Cremella, Italy) (UV-Vis). All thicknesses were measured using a Bruker Profilometer (Billerica, MA, USA).

### 3. Results and Discussion

We initially tried to coat the substrates for spin and spray coatings using 10%-by-weight solutions containing only the dyes. However, this approach has been discarded due to the low optical quality and poor reproducibility of the obtained films. To obtain better quality coatings, we decide to use the polymeric hosts. The host–guest approach with an organic dye embedded in a suitable polymeric host is very diffused, especially for optoelectronic applications [24,25]. It allows us to maintain the optical characteristics of the dye, in our case UV shielding, by combining them with the ability to form films of good optical and mechanical quality imparted by the polymeric matrix. In order to use this approach, it is necessary that the host and the guest have good chemical compatibility. That is, the dye disperses well in the polymer and does not give rise to aggregation phenomena that could modify its optical and mechanical characteristics. In the case of quinine, since it is soluble in water, the PVA is selected as a polymeric host. Indeed, the PVA is also soluble in water and with the requisites of suitability for contact with food. Unfortunately, the use of water as a solvent leads to poor-quality spin-coated films with inhomogeneous coverage. For this reason, we decide to not use this technique for the PVA-Q host–guest combination.

Since p-Coumaric acid is soluble in alcohol, a derivative of polyvinyl butyrate (Mowital) is chosen as a polymeric host because of its ability to dissolve in ethanol but not in water. Moreover, Mowital has been shown to possess biocompatibility and non-toxicity [25–27]. Thanks to the presence of the polymeric hosts, films of good quality are obtained for spin coating for the host–guest combination PVB-PCA and for both host–guest systems using the spray-coating technique. It is worth noting that spray coating is a low-cost technique that can be applied to coat surfaces of large dimensions and strange shapes. It can also be applied to ready-made packaging, as well as to C or PLA preformed film without modifying the preparation procedures.

To verify the deposition on the substrates, systematic SEM characterizations are carried out. Figure 2 shows a comparison of the morphologies of the cellulose substrates spray-coated with the two blends. We look to the PVB-PCA and the PVA-Q films. For all films, large-scale surfaces are covered and no cracks are observed. The circular defect, surrounded by the white circle in Figure 2b, is a slight concavity, a few nanometers deep. It is already present in the initial cellulose substrate supplied by Corapack Srl, as confirmed also by the AFM image (see the white circle in Figure 3a) where we can distinguish this small concavity. We deduce that they are due to the commercial production and do not alter the quality of the cellulose. The contaminations instead (see the one surrounded by the black circle in Figure 2b) appear with a lighter contrast and consist of chemical particulates not perfectly dissolved in solution.

No differences in homogeneity distribution are observed with SEM characterizations. When the SEM contrast of a non-conductive sample is interpreted, the charging effect from electron/ion irradiation is unavoidable. In general, the charge effect is caused by the accumulation of static electric charges on the surface of the sample: in particular, this effect is very strong on the PLA films, making a SEM characterization of the coatings on this substrate impossible. Therefore, for these films, only the AFM technique is used to study their morphology and roughness (Figure 3d–f). Accurate width and height measurements have required that the measured width be corrected for AFM tip convolution effects. This is accomplished by using an internal calibration standard (e.g., gold nanoparticles, AuNPs) for in situ calibration of the AFM tip radius and to monitor changes in tip size [28]. All AFM images are processed with align rows-median to remove skipping lines.

In the case of cellulose, the initial roughness of the film (Figure 3a) is greater than 5 nm and increases between 6 and 9 nm after deposition of the PVA-Q (Figure 3b) and PVB-PCA (Figure 3c) by spray coating. Both deposition processes fail to deposit a perfectly flat film and the formation of nanostructures deriving from the solutions is observed. However, this nano-roughness is systematic on the whole sample and, as reported below, does not alter the optical properties of the coating. Indeed, the original rms for the PLA film is about 1 nm (Figure 3d), while the roughness of the same film after the deposition by spray coating

and spin coating of the blend PVB-PCA does not increase, resulting in an average of 2 nm (Figure 3e,f, respectively). In conclusion, both substrates do not change their morphology in a dramatic way after the deposition of the two polymeric blends.

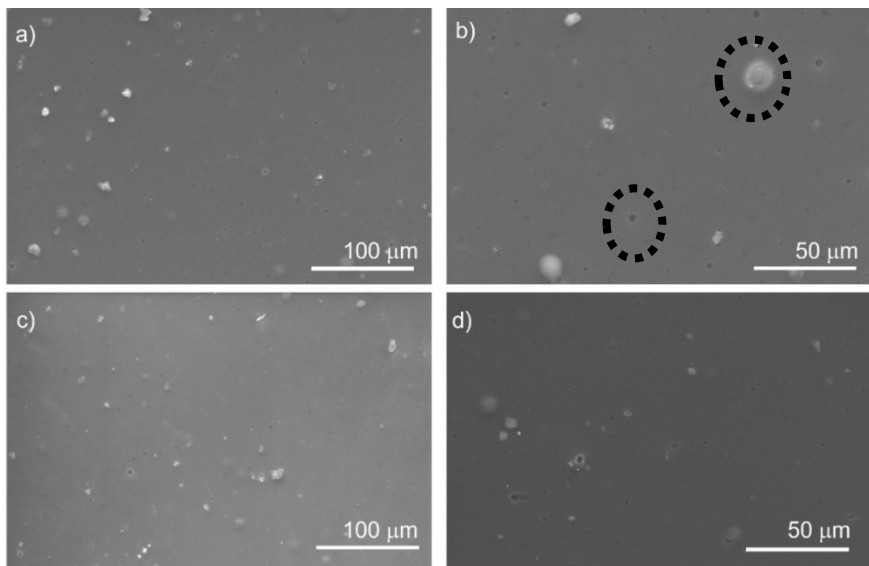

**Figure 2.** SEM planar-view images of cellulose substrates coated with the two blends. The holes observed in all images are due to imperfections already present in the initial Corapack cellulose film. (**a**,**b**) SEM images at different magnifications of PVB-PCA obtained by spray coating; (**c**,**d**) SEM images obtained in different areas of the sample after a spray-coating deposition of PVA-Q. All surfaces are homogeneous with randomly distributed granulates that derive from the imperfect dissolution of the substances in the initial solvent.

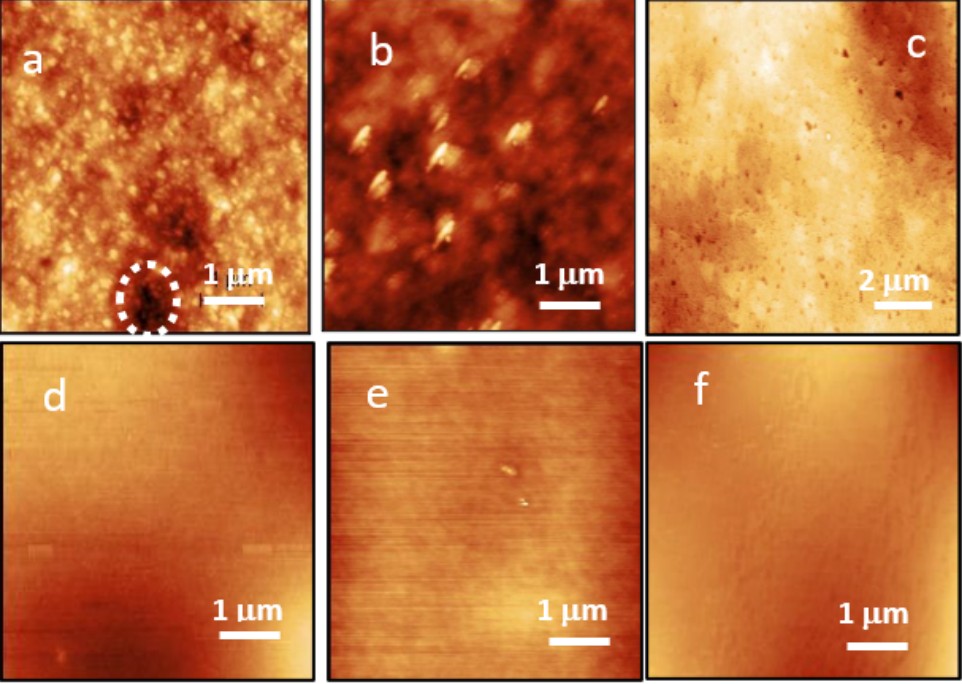

**Figure 3.** Tapping mode AFM images: (**a**) Corapack's cellulose substrate, with the white circle surrounding the small fabrication defect. Deposited by spray coating on cellulose: (**b**) PVA-Q; (**c**) PVB-PCA; (**d**) Corapack's PLA substrate. Deposited on PLA: (**e**) PVB-PCA deposited by spray coating; (**f**) PVB-PCA deposited by spin coating.

*Spectroscopical Analysis*

Figure 4 shows the transmission spectra of the PVA-Quinine blend (Figure 4a,c) and the PVB-P Coumaric acid blend (Figure 4b,d) on the PLA (Figure 4a,b) and cellulose substrates (Figure 4c,d). In Figure 4a,b, the black solid lines represent the transmission spectrum of the PLA substrate, while in Figure 4c,d they show the transmission spectrum of the cellulose substrate. The plots show that the PLA substrate transmits at least 80% of light in the spectrum between 600 nm and 250 nm, while the cellulose substrate transmits at least 70%. From the same initial solution, the blend PVA-Quinine is spray-coated on the different substrates. It is clear that the light transmission in the UV region from 350 nm to 250 nm drops to almost 0% on the PLA substrate (Figure 4a) and to 10% on the cellulose substrate (Figure 4c). The quinine molecule does not adhere to the two substrates in the same way, preferring the PLA; in fact, the UV barrier is around 100% when the substrate is PLA instead of cellulose. Figure 4b,d shows the transmission spectra for the p-Coumaric blend on the two substrates deposited with the spin- (red lines) and spray-coating (black line) technique. Even here, the UV barrier is higher using the PLA substrate instead of the cellulose. However, for the Coumaric blend, the difference of the transmission spectra between the two substrates is not as significant. For the spray-coating films, the transmission of the UV light (250–350 nm spectral region) is so low that the instrument is unable to measure it. This is a strong indication that the spray-coating technique is very useful in covering large substrates, even if it is more difficult to control the thickness of the coatings. Using the transmission values, we can also estimate the thickness ratio between the spray and spin-coated films on the different substrates.

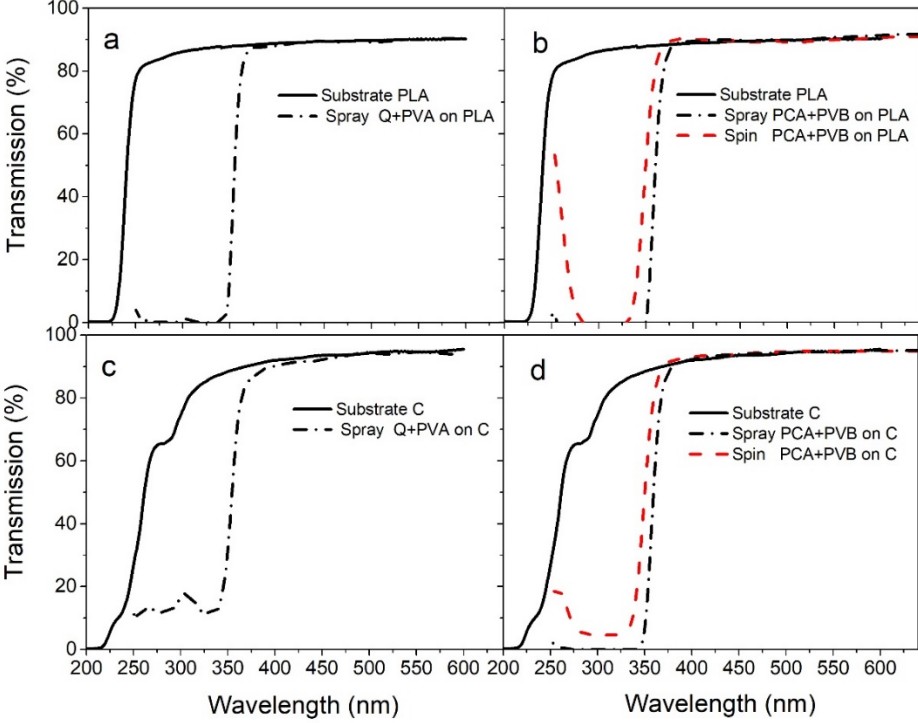

**Figure 4.** (**a**) Transmission spectra of the PLA substrate (black solid line) and of the spray-coated film of the blend PVA-Quinine on PLA substrate (black dash line); (**b**) Transmission spectra of the PLA substrate (black solid line) and of the spray-coated film (black dash line) and the spin-coated film (red dash line) of the blend PVB-P Coumaric acid on PLA substrate; (**c**) Transmission spectra of the cellulose substrate (black solid line) and of the spray-coated film of the blend PVA-Quinine on the cellulose substrate (**d**) Transmission spectra of the cellulose substrate (black solid line) and of the spray-coated film (black dash line) and the spin-coated film (red dashed line) of the blend PVB-P Coumaric acid on cellulose substrate.

We know that:

$$\frac{T}{T_0}(\lambda) = e^{-\alpha(\lambda)d} \tag{1}$$

where $T/T_0(\lambda)$ is the fraction of the light transmitted through the sample at a given wavelength, $\alpha(\lambda)$ is the absorption coefficient at a given wavelength, and $d$ is the thickness of the film. Moreover, $\alpha(\lambda)$ is equal to $n^*\sigma(\lambda)$, where n is the density of molecule per unit volume and $\sigma$ is the absorption cross-section of a single molecule at a given wavelength. From Equation (1), we can calculate the thickness d using the indirect formula:

$$d = \frac{\ln\left(\frac{T}{T_o}(\lambda)\right)}{\alpha(\lambda)} \tag{2}$$

Considering the same $\alpha$ (350 nm), due to the fact that the same molecule dispersed in the same matrix has the same absorption coefficient $\alpha(\lambda)$, we can estimate that the spray-coated film is more than three times thicker than the spin-coated film on the two substrates.

Since the initial substrates of PLA and C are extremely soft and flexible, in order to have an indication of the thickness of the films deposited through the two different approaches, the Mowital-based mixtures have been deposited on a glass by spin and spray coating. The samples are then measured with a profilometer. The thicknesses obtained are around 80 nm ($\pm$10 nm) for spin coating and 250 nm ($\pm$25 nm) for spray coating. The ratio found on glass samples of about 1:3 is in excellent agreement with the ratio obtained from the transmission data, confirming that the spray-coating technique deposits film three times thicker than the same material deposited by spin coating.

## 4. Conclusions

In this paper, we show the possibility of reducing the UV transparency of two commercial and very common food packages after smart coating with commercial and green materials such as p-Coumaric acid and quinine. We use different coating techniques such as spray and spin coating to show that it is possible to deposit the molecules in a fast way, largely increasing the UV shielding. It is worth noting that spray coating is a low-cost technique that allows surfaces of different shapes and sizes to be coated, and it can also be applied to ready-made packaging. The films are studied spectroscopically and morphologically, revealing that even if the substrates are not modified by the coatings, the UV barrier is increased by at least 70%. In this way, it is possible to predict the packaging's future use in the market.

**Author Contributions:** Conceptualization, T.V., M.P. and M.B.; methodology, T.V., M.P., M.B., M.G. and N.T.; validation, T.V., M.P. and M.B.; writing—original draft preparation, T.V., M.P. and M.B.; writing—review and editing, T.V., M.P. and M.B. All authors have read and agreed to the published version of the manuscript.

**Funding:** This work was supported by the sPATIALS3 project financed by (ERDF ROP) 2014–2020—FESR Regione Lombardia—Axis I: "Strengthen technological research, development and innovation (RD&I)"—Action 1.b.1.3: "Support for co-operative R&D activities to develop new sustainable technologies, products and services"—Call Hub.

**Institutional Review Board Statement:** Not applicable.

**Informed Consent Statement:** Not applicable.

**Data Availability Statement:** The data presented in this study are available on request from the corresponding author. The data are not publicly available due to privacy.

**Conflicts of Interest:** The authors declare no conflict of interest.

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
