# Peer review of "Natural Dyes Used as Organic Coatings UV Protecting for Food Packages"

_coatings, doi:10.3390/coatings12030417_

Round 1
Reviewer 1 Report
1- title is not comprehensive (pls. refer to base materials)
2-some of the references are not related , because their structures are nanostructures.
3-uv protections are not clear
what about other properties?
in morphological characterization ( in all images you can find a few defects what are they ?
how is the effect of thickness ?
please make conclusion more clear
you have referred spin coated twice why?
Reviewer 2 Report
The article entitled ‘’Organic Coatings UV Protecting for Food Packages’’ , it is evident that a research field is of great interest, especially in the last few years. Therefore, before publishing, I suggest making some changes:
- a graphical abstract would be very helpful for better understanding the article. Please insert one.
- the introductory part needs improvements in other people's research on this topic in the last 2 years, many citations are older even from 2008.
- Authors highlighted that food packaging is used to protect food from environmental contamination, This is correct; however, they forget to mention that active packages are also used to extend the shelf life of products which are reported in the literature and well described: See for examples: doi:10.3390/polym12081748, https://doi.org/10.3390/foods10020401
- At the line 58 please mention the exactly concentration of the polyvinyl alcohol (the Mw)
- the results are well described
Reviewer 3 Report
This manuscript shows a potential way that can overcome the low-UV protecting ability of commercially used PLA plastics. This was achieved through spin and spray coatings of PLA plastics with commercial dyes (e.g. Quinine, P-Coumatic acid). The successful coating was then demonstrated by following characterization with AFM and SEM tools. In general, I think the manuscript is publishable, however, with the following concerns being addressed.
- The authors should better estimate the additional costs with the current deposition technique. This will show the significance of the manuscript.
- Lines 109~131. There are too many discussions while little data was provided to support the discussion. For instance, can authors provide certain data to show the the optical quality of original plastic and the coated plastics?
- I haven concerns on the stability of this coating. Are there any data to show that the coatings are stable enough and will not fall off?
Reviewer 4 Report
The content of this paper was to report the technique to develop the organic coatings. However, the research was not finish. No practical experiment to proof the UV protecting and humidity resistance for food packages.
The function of the UV protecting and humidity resistance was only described:
In abstract, authors mentioned that “Our result shows that it is possible to enhance the UV protection of the packaging using a suitable deposition of these biocompatible materials.”
In conclusion, authors proposed that “in this way it is possible to predict their future use in the market. In particular since the coumaric based films are not soluble in water this ensures their resistance even in conditions of high humidity such as in the case of contact with fresh food.”
Please perform these performance test, provide these information and rewrote the paper.
Round 2
Reviewer 1 Report
how do you control the thickness of coating and it"s effect
I am not satisfied with your answer about thickness of coating.
Reviewer 3 Report
I think the manuscript can be accepted in the present form.
Reviewer 4 Report
The content of revised manuscript have been improved significantly. Authors made such statement.
“In this way it is possible to predict their future use in the market. In particular since the coumaric-based films are not soluble in water this could ensure their resistance even in conditions of high humidity such as in the case of contact with fresh food.”
Please provide more evidence or statement to illustrate the possible utilization.
Author Response
Answer: We understand that this sentence can still be misunderstood. So we decided to eliminate it, because as we said also in the previous reply the test of humidity resistance is beyond the aim of our paper.